# Exponential Family Estimation via Adversarial Dynamics Embedding

*Bo Dai[1], *Zhen Liu[2], *Hanjun Dai[1], Niao He[3], Arthur Gretton[4], Le Song[5,6], Dale Schuurmans[1,7]

[1]Google Research, Brain Team, [2]Mila, University of Montreal,
[3]University of Illinois at Urbana Champaign, [4]University College London,
[5]Georgia Institute of Technology, [6]Ant Financial, [7]University of Alberta

## Abstract

We present an efficient algorithm for maximum likelihood estimation (MLE) of exponential family models, with a general parametrization of the energy function that includes neural networks. We exploit the primal-dual view of the MLE with a *kinetics augmented model* to obtain an estimate associated with an *adversarial* dual sampler. To represent this sampler, we introduce a novel neural architecture, *dynamics embedding*, that generalizes Hamiltonian Monte-Carlo (HMC). The proposed approach inherits the flexibility of HMC while enabling tractable entropy estimation for the augmented model. By learning both a dual sampler and the primal model simultaneously, and sharing parameters between them, we obviate the requirement to design a separate sampling procedure once the model has been trained, leading to more effective learning. We show that many existing estimators, such as contrastive divergence, pseudo/composite-likelihood, score matching, minimum Stein discrepancy estimator, non-local contrastive objectives, noise-contrastive estimation, and minimum probability flow, are special cases of the proposed approach, each expressed by a different (fixed) dual sampler. An empirical investigation shows that adapting the sampler during MLE can significantly improve on state-of-the-art estimators[1].

## 1  Introduction

The exponential family is one of the most important classes of distributions in statistics and machine learning, encompassing undirected graphical models (Wainwright and Jordan, 2008) and energy-based models (LeCun et al., 2006; Wu et al., 2018), which include, for example, Markov random fields (Kinderman and Snell, 1980), conditional random fields (Lafferty et al., 2001) and language models (Mnih and Teh, 2012). Despite the flexibility of this family and the many useful properties it possesses (Brown, 1986), most such distributions are intractable because the partition function does not possess an analytic form. This leads to difficulty in evaluating, sampling and learning exponential family models, hindering their application in practice. In this paper, we consider a longstanding question:

> *Can a simple yet effective algorithm be developed for estimating general exponential family distributions?*

There has been extensive prior work addressing this question. Many approaches focus on approximating maximum likelihood estimation (MLE), since it is well studied and known to possess desirable statistical properties, such as consistency, asymptotic unbiasedness, and asymptotic normality (Brown, 1986). One prominent example is contrastive divergence (CD) (Hinton, 2002) and its variants (Tieleman and Hinton, 2009; Du and Mordatch, 2019). It approximates the gradient of the log-likelihood by a stochastic estimator that uses samples generated from a few Markov chain Monte Carlo (MCMC) steps. This approach has two shortcomings: first and foremost, the stochastic gradient is *biased*,

which can lead to poor estimates; second, CD and its variants require careful design of the MCMC transition kernel, which can be challenging.

Given these difficulties with MLE, numerous learning criteria have been proposed to avoid the partition function. Pseudo-likelihood estimators (Besag, 1975) approximate the joint distribution by the product of conditional distributions, each of which only represents the distribution of a single random variable conditioned on the others. However, the the partition function of each factor is still generally intractable. Score matching (Hyvärinen, 2005) minimizes the Fisher divergence between the empirical distribution and the model. Unfortunately, it requires third order derivatives for optimization, which becomes prohibitive for large models (Kingma and LeCun, 2010; Li et al., 2019). Noise-contrastive estimation (Gutmann and Hyvärinen, 2010) recasts the problem as ratio estimation between the target distribution and a pre-defined auxiliary distribution. However, the auxiliary distribution must cover the support of the data with an analytical expression that still allows efficient sampling; this requirement is difficult to satisfy in practice, particularly in high dimensional settings. Minimum probability flow (Sohl-Dickstein et al., 2011) exploits the observation that, ideally, the empirical distribution will be the stationary distribution of transition dynamics defined under an optimal model. The model can then be estimated by matching these two distributions. Even though this idea is inspiring, it is challenging to construct appropriate dynamics that yield efficient learning.

In this paper, we introduce a novel algorithm, *Adversarial Dynamics Embedding (ADE)*, that directly approximates the MLE while achieving computational and statistical efficiency. Our development starts with the *primal-dual* view of the MLE (Dai et al., 2019) that provides a natural objective for jointly learning both a sampler and a model, as a remedy for the expensive and biased MCMC steps in the CD algorithm. To parameterize the dual distribution, Dai et al. (2019) applies a naive transport mapping, which makes entropy estimation difficult and requires learning an extra auxiliary model, incurring additional computational and memory cost.

We overcome these shortcomings by considering a different approach, inspired by the properties of Hamiltonian Monte-Carlo (HMC) (Neal, 2011):

   **i)** HMC forms a stationary distribution with *independent* potential and kinetic variables;
   **ii)** HMC can approximate the exponential family *arbitrarily closely*.

As in HMC, we consider an *augmented model* with latent kinetic variables in Section 3.1, and introduce a novel neural architecture in Section 3.2, called *dynamics embedding*, that mimics sampling and represents the dual distribution via parameters of the primal model. This approach shares with HMC the advantage of a *tractable* entropy function for the augmented model, while enriching the flexibility of sampler without introducing extra parameters. In Section 3.3 we develop a max-min objective that allows the shared parameters in primal model and dual sampler to be learned simultaneously, which improves computational and sample efficiency. We further show that the proposed estimator subsumes CD, pseudo-likelihood, score matching, non-local contrastive objectives, noise-contrastive estimation, and minimum probability flow as special cases with hand-designed dual samplers in Section 4. Finally, in Section 5 we find that the proposed approach can outperform current state-of-the-art estimators in a series of experiments.

## 2   Preliminaries

**Exponential family and energy-based model**   The natural form of the exponential family over $\Omega \subset \mathbb{R}^d$ is defined as

$$p_{f'}(x) = \exp\left(f'(x) - \log p_0(x) - A_{p_0}(f')\right), \ A_{p_0}(f') := \log \int_\Omega \exp\left(f'(x)\right) p_0(x)\, dx, \quad (1)$$

where $f'(x) = w^\top \phi_\varpi(x)$. The sufficient statistic $\phi_\varpi(\cdot) : \Omega \to \mathbb{R}^k$ can be any general parametric model, *e.g.*, a neural network. The $(w, \varpi)$ are the parameters to be learned from observed data. The exponential family definition (1) includes the energy-based model (LeCun et al., 2006) as a special case, by setting $f'(x) = \phi_\varpi(x)$ with $k = 1$, which has been generalized to the infinite dimensional case (Sriperumbudur et al., 2017). The $p_0(x)$ is fixed and covers the support $\Omega$, which is usually unknown in practical high-dimensional problems. Therefore, we focus on learning $f(x) = f'(x) - \log p_0(x)$ jointly with $p_0(x)$, which is more difficult: in particular, the doubly dual embedding approach (Dai et al., 2019) is no longer applicable.

Given a sample $\mathcal{D} = [x_i]_{i=1}^N$ and denoting $f \in \mathcal{F}$ as the valid parametrization family, an exponential family model can be estimated by maximum log-likelihood, *i.e.*,

$$\max_{f \in \mathcal{F}} \ L(f) := \widehat{\mathbb{E}}_\mathcal{D}[f(x)] - A(f), \ A(f) = \log \int_\Omega \exp\left(f(x)\right) dx, \quad (2)$$

with gradient $\nabla_f L(f) = \widehat{\mathbb{E}}_{\mathcal{D}}[\nabla_f f(x)] - \mathbb{E}_{p_f(x)}[\nabla_f f(x)]$. Since $A(f)$ and $\mathbb{E}_{p_f(x)}[\nabla_f f(x)]$ are both intractable, solving the MLE for a general exponential family model is very difficult.

**Dynamics-based MCMC**  Dynamics-based MCMC is a general and effective tool for sampling. The idea is to represent the target distribution as the solution to a set of (stochastic) differential equations, which allows samples from the target distribution to be obtained by simulating along the dynamics defined by the differential equations.

HMC (Neal, 2011) is a representative algorithm in this category, which exploits the well-known Hamiltonian dynamics. Specifically, given a target distribution $p_f(x) \propto \exp(f(x))$, the Hamiltonian is defined as $\mathcal{H}(x,v) = -f(x) + k(v)$, where $k(v) = \frac{1}{2}v^\top v$ is the kinetic energy. The Hamiltonian dynamics generate $(x,v)$ over time $t$ by following

$$\left[\frac{dx}{dt}, \frac{dv}{dt}\right] = [\partial_v \mathcal{H}(x,v), -\partial_x \mathcal{H}(x,v)] = [v, \nabla_x f(x)]. \tag{3}$$

Asymptotically as $t \to \infty$, $x$ visits the underlying space according to the target distribution. In practice, to reduce discretization error, an acceptance-rejection step is introduced. The finite-step dynamics-based MCMC sampler can be used for approximating $\mathbb{E}_{p_f(x)}[\nabla_f f(x)]$ in $\nabla_f L(f)$, which leads to the CD algorithm (Hinton, 2002; Zhu and Mumford, 1998).

**Primal-dual view of MLE**  The Fenchel duality of $A(f)$ has been exploited (Rockafellar, 1970; Wainwright and Jordan, 2008; Dai et al., 2019) as another way to address the intractability of the log-partition function.

**Theorem 1 (Fenchel dual of log-partition (Wainwright and Jordan, 2008))** *Let* $H(q) := -\int_\Omega q(x) \log q(x) dx$. *Then:*

$$A(f) = \max_{q \in \mathcal{P}} \langle q(x), f(x) \rangle + H(q), \quad p_f(x) = \operatorname{argmax}_{q \in \mathcal{P}} \langle q(x), f(x) \rangle + H(q), \tag{4}$$

*where $\mathcal{P}$ denotes the space of distributions and $\langle f, g \rangle = \int_\Omega f(x) g(x) dx$.*

Plugging the Fenchel dual of $A(f)$ into the MLE (2), we arrive at a $\max$-$\min$ reformulation

$$\max_{f \in \mathcal{F}} \min_{q \in \mathcal{P}} \widehat{\mathbb{E}}_{\mathcal{D}}[f(x)] - \mathbb{E}_{q(x)}[f(x)] - H(q), \tag{5}$$

which bypasses the explicit computation of the partition function. Another byproduct of the primal-dual view is that the dual distribution can be used for inference, however in vanila estimators this usually requires expensive sampling algorithms.

The dual sampler $q(\cdot)$ plays a vital role in the primal-dual formulation of the MLE in (5). To achieve better performance, we have several principal requirements in parameterizing the dual distribution:

    **i)** the parametrization family needs to be *flexible* enough to achieve small error in solving the inner minimization problem;
    **ii)** the entropy of the parametrized dual distribution should be *tractable*.

Moreover, as shown in (4) in Theorem 1, the optimal dual sampler $q(\cdot)$ is determined by primal potential function $f(\cdot)$. This leads to the third requirement:

    **iii)** the parametrized dual sampler should *explicitly incorporate* the primal model $f$.

Such a dependence can potentially reduce both the memory and learning sample complexity.

A variety of techniques have been developed for distribution parameterization, such as reparametrized latent variable models (Kingma and Welling, 2014; Rezende et al., 2014), transport mapping (Goodfellow et al., 2014), and normalizing flow (Rezende and Mohamed, 2015; Dinh et al., 2017; Kingma et al., 2016). However, none of these satisfies the requirements of flexibility and a tractable density simultaneously, nor do they offer a principled way to couple the parameters of the dual sampler with the primal model.

## 3   Adversarial Dynamics Embedding

By augmenting the original exponential family with kinetic variables, we can parametrize the dual sampler with a *dynamics embedding* that satisfies all three requirements without effecting the MLE, allowing the primal potential function and dual sampler to both be trained adversarially. We start with the embedding of classical Hamiltonian dynamics (Neal, 2011; Caterini et al., 2018) for the dual sampler parametrization, as a concrete example, then discuss its generalization in latent space and the stochastic Langevin dynamics embedding. This technique is extended to other dynamics, with their own advantages, in Appendix B.

## 3.1 Primal-Dual View of Augmented MLE

As noted, it is difficult to find a parametrization of $q(x)$ in (5) that simultaneously satisfies all three requirements. Therefore, instead of directly tackling (5) in the original model, and inspired by HMC, we consider the augmented exponential family $p(x, v)$ with an auxiliary momentum variable, *i.e.*,

$$p(x, v) = \frac{\exp\left(f(x) - \frac{\lambda}{2} v^\top v\right)}{Z(f)}, \quad Z(f) = \int \exp\left(f(x) - \frac{\lambda}{2} v^\top v\right) dx dv. \tag{6}$$

The MLE of such a model can be formulated as

$$\max_f L(f) := \widehat{\mathbb{E}}_{x \sim \mathcal{D}} \left[\log \int p(x, v) \, dv\right] = \widehat{\mathbb{E}}_{x \sim \mathcal{D}} \mathbb{E}_{p(v|x)} \left[f(x) - \frac{\lambda}{2} v^\top v - \log p(v|x)\right] - \log Z(f) \tag{7}$$

where the last equation comes from true posterior $p(v|x) = \mathcal{N}\left(0, \lambda^{-\frac{1}{2}} I\right)$ due to the independence of $x$ and $v$. This independence also induces the equivalent MLE as proved in Appendix A.

**Theorem 2 (Equivalent MLE)** *The MLE of the augmented model is the same as the original MLE.*

Applying the Fenchel dual to $Z(f)$ of the augmented model (6), we derive a primal-dual formulation of (7), leading to the objective,

$$L(f) \propto \min_{q(x,v) \in \mathcal{P}} \widehat{\mathbb{E}}_{x \sim \mathcal{D}} [f(x)] - \mathbb{E}_{q(x,v)} \left[f(x) - \frac{\lambda}{2} v^\top v - \log q(x, v)\right]. \tag{8}$$

The $q(x, v)$ in (8) contains momentum $v$ as the latent variable. One can also exploit the latent variable model for $q(x) = \int q(x|v) q(v) \, dv$ in (5). However, the $H(q)$ in (5) requires marginalization, which is intractable in general, and usually estimated through variational inference with the introduction of an extra posterior model $q(v|x)$. Instead, by considering the specifically designed augmented model, (8) eliminates these extra variational steps.

Similarly, one can consider the latent variable augmented model with multiple momenta, *i.e.*,
$p\left(x, \{v^i\}_{i=1}^T\right) = \frac{\exp\left(f(x) - \sum_{i=1}^T \frac{\lambda_i}{2} \|v^i\|_2^2\right)}{Z(f)}$, leading to the optimization

$$L(f) \propto \min_{q\left(x, \{v^i\}_{i=1}^T\right) \in \mathcal{P}} \widehat{\mathbb{E}}_{x \sim \mathcal{D}} [f(x)] - \mathbb{E}_{q\left(x, \{v^i\}_{i=1}^T\right)} \left[f(x) - \sum_{i=1}^T \frac{\lambda_i}{2} \|v^i\|_2^2 - \log q\left(x, \{v^i\}_{i=1}^T\right)\right]. \tag{9}$$

## 3.2 Representing Dual Sampler via Primal Model

We now introduce the Hamiltonian dynamics embedding to represent the dual sampler $q(\cdot)$, as well as its generalization and special instantiation that satisfy all three of the principal requirements.

The vanilla HMC is derived by discretizing the Hamiltonian dynamics (3) with a leapfrog integrator. Specifically, in a single time step, the sample $(x, v)$ moves towards $(x', v')$ according to

$$(x', v') = \mathbf{L}_{f,\eta}(x, v) := \begin{pmatrix} v^{\frac{1}{2}} = v + \frac{\eta}{2} \nabla_x f(x) \\ x' = x + \eta v^{\frac{1}{2}} \\ v' = v^{\frac{1}{2}} + \frac{\eta}{2} \nabla_x f(x') \end{pmatrix}, \tag{10}$$

where $\eta$ is defined as the leapfrog stepsize. Let us denote the one-step leapfrog as $(x', v') = \mathbf{L}_{f,\eta}(x, v)$ and assume the $(x^0, v^0) \sim q_\theta^0(x, v)$. After $T$ iterations, we obtain

$$\left(x^T, v^T\right) = \mathbf{L}_{f,\eta} \circ \mathbf{L}_{f,\eta} \circ \ldots \circ \mathbf{L}_{f,\eta} \left(x^0, v^0\right). \tag{11}$$

Note that this can be viewed as a neural network with a special architecture, which we term *Hamiltonian (HMC) dynamics embedding*. Such a representation explicitly characterizes the dual sampler by the primal model, *i.e.*, the potential function $f$, meeting the dependence requirement.

The flexibility of the distributions HMC embedding actually is ensured by the nature of the dynamics-based samplers. In the limiting case, the proposed neural network (11) reduces to a gradient flow, whose stationary distribution is exactly the model distribution:

$$p(x, v) = \text{argmax}_{q(x,v) \in \mathcal{P}} \mathbb{E}_{q(x,v)} \left[f(x) - \frac{\lambda}{2} v^\top v - \log q(x, v)\right].$$

The approximation strength of the HMC embedding is formally justified as follows:

**Theorem 3 (HMC embeddings as gradient flow)** *In continuous time,* i.e. *with infinitesimal step-size $\eta \to 0$, the density of particles $(x^t, v^t)$, denoted $q^t(x, v)$, follows the Fokker-Planck equation*

$$\frac{\partial q^t(x,v)}{\partial t} = \nabla \cdot \left(q^t(x, v) G \nabla \mathcal{H}(x, v)\right), \tag{12}$$

*with $G = \begin{bmatrix} 0 & \mathbf{I} \\ -\mathbf{I} & 0 \end{bmatrix}$, which has a stationary distribution $p(x, v) \propto \exp(-\mathcal{H}(x, v))$ with the marginal distribution $p(x) \propto \exp(f(x))$.*

Details of the proofs are given in Appendix A. Note that this stationary distribution result is an instance of the more general dynamics described in Ma et al. (2015), showing the flexility of the induced distributions. As demonstrated in Theorem 3, the neural parametrization formed by the HMC embedding is able to well approximate an exponential family distribution on continuous variables.

**Remark (Generalized HMC dynamics in latent space)** The leapfrog operation in vanilla HMC works directly in the original observation space, which could be high-dimensional and noisy. We generalize the leapfrog update rule to the latent space and form a new dynamics as follows,

$$(x', v') = \mathbf{L}_{f,\eta,S,g}(x,v) := \begin{pmatrix} v^{\frac{1}{2}} = v \odot \exp\left(S_v\left(\nabla_x f(x), x\right)\right) + \frac{\eta}{2} g_v\left(\nabla_x f(x), x\right) \\ x' = x \odot \exp\left(S_x\left(v^{\frac{1}{2}}\right)\right) + \eta g_x\left(v^{\frac{1}{2}}\right) \\ v' = v^{\frac{1}{2}} \odot \exp\left(S_v\left(\nabla_x f(x'), x'\right)\right) + \frac{\eta}{2} g_v\left(\nabla_x f(x'), x'\right) \end{pmatrix}, \qquad (13)$$

where $v \in \mathbb{R}^l$ denote the momentum evolving space and $\odot$ denotes element-wise product. Specifically, the terms $S_v\left(\nabla_x f(x), x\right)$ and $S_x\left(v^{\frac{1}{2}}\right)$ rescale $v$ and $x$ coordinatewise. The term $g_v\left(\nabla_x f(x), x\right) \mapsto \mathbb{R}^l$ can be understood as projecting the gradient information to the essential latent space where the momentum is evolving. Then, for updating $x$, the latent momentum is projected back to original space via $g_x\left(v^{\frac{1}{2}}\right) \mapsto \Omega$. With these generalized leapfrog updates, the dynamical system avoids operating in the high-dimensional noisy input space, and becomes more computationally efficient. We emphasize that the proposed generalized leapfrog parametrization (13) is different from the one used in Levy et al. (2018), which is inspired from the real-NVP flow (Dinh et al., 2017).

By the generalized HMC embedding (13), we have a flexible layer $(x', v') = \mathbf{L}_{f,\eta,S,g}(x,v)$, where $(S_v, S_x, g_v, g_x)$ will be learned in addition to the stepsize. Obviously, the classic HMC layer $\mathbf{L}_{f,\eta,M}(x,v)$ is a special case of $\mathbf{L}_{f,\eta,S,g}(x,v)$ by setting $(S_v, S_x)$ to zero and $(g_v, g_f)$ to identity functions.

**Remark (Stochastic Langevin dynamics)** The stochastic Langevin dynamics can also be recovered from the leapfrog step by resampling momentum in every step. Specifically, the sample $(x, \xi)$ moves according to

$$(x', v') = \mathbf{L}_{f,\eta}^{\xi}(x) := \begin{pmatrix} v' = \xi + \frac{\eta}{2} \nabla_x f(x) \\ x' = x + v' \end{pmatrix}, \text{ with } \xi \sim q_\theta(\xi). \qquad (14)$$

Hence, stochastic Langevin dynamics resample $\xi$ to replace the momentum in leapfrog (10), ignoring the accumulated gradients. By unfolding $T$ updates, we obtain

$$\left(x^T, \{v^i\}_{i=1}^T\right) = \mathbf{L}_{f,\eta}^{\xi^{T-1}} \circ \mathbf{L}_{f,\eta}^{\xi^{T-2}} \circ \ldots \circ \mathbf{L}_{f,\eta}^{\xi^0}(x^0) \qquad (15)$$

as the derived neural network. Similarly, we can also generalize the stochastic Langevin updates $\mathbf{L}_{f,\eta}^{\xi}$ to a low-dimension latent space by introducing $g_v\left(\nabla_x f(x), x\right)$ and $g_x(v')$ correspondingly.

One of the major advantages of the proposed distribution parametrization is its density value is also tractable, leading to tractable entropy estimation in (8) and (9). In particular, we have the following,

**Theorem 4 (Density value evaluation)** *If $\left(x^0, v^0\right) \sim q_\theta^0(x,v)$, after $T$ vanilla HMC steps (10), then*

$$q^T\left(x^T, v^T\right) = q_\theta^0\left(x^0, v^0\right). \qquad (16)$$

*For $\left(x^T, v^T\right)$ from the generalized leapfrog steps (13), we have*

$$q^T\left(x^T, v^T\right) = q_\theta^0\left(x^0, v^0\right) \prod_{t=1}^T \left(\Delta_x(x^t) \Delta_v(v^t)\right), \qquad (17)$$

*where $\Delta_x(x^t)$ and $\Delta_v(v^t)$ denote*

$$\Delta_x(x^t) = \left|\det\left(\text{diag}\left(\exp\left(2S_v\left(\nabla_x f(x^t), x^t\right)\right)\right)\right)\right|, \Delta_v(v^t) = \left|\det\left(\text{diag}\left(\exp\left(S_x\left(v^{\frac{1}{2}}\right)\right)\right)\right)\right|. \qquad (18)$$

*For $\left(x^T, \{v^i\}_{i=1}^T\right)$ from the Langevin dynamics (14) with $\left(x^0, \{\xi^i\}_{i=0}^{T-1}\right) \sim q_\theta^0(x, \xi) \prod_{i=i}^{T-1} q_{\theta_i}(\xi)$, we have*

$$q^T\left(x^T, \{v^i\}_{i=1}^T\right) = q_\theta^0\left(x^0, \xi^0\right) \prod_{i=1}^{T-1} q_{\theta_i}(\xi^i). \qquad (19)$$

The proof of Theorem 4 can be found in Appendix A.

The proposed dynamics embedding satisfies all three requirements: it defines a flexible family of distributions with computable entropy; and couples the learning of the dual sampler with the primal model, leading to memory and sample efficient learning algorithms, as we introduce in next section.

## 3.3 Coupled Model and Sampler Learning

By plugging the $T$-step Hamiltonian dynamics embedding (10) into the primal-dual MLE of the augmented model (8) and applying the density value evaluation (16), we obtain the proposed optimization, which learns primal potential $f$ and the dual sampler adversarially,

$$\max_{f \in \mathcal{F}} \min_{\Theta} \ell(f, \Theta) := \widehat{\mathbb{E}}_{\mathcal{D}}[f] - \mathbb{E}_{(x^0, v^0) \sim q_\theta^0(x,v)} \left[ f\left(x^T\right) - \frac{\lambda}{2} \left\| v^T \right\|_2^2 \right] - H\left(q_\theta^0\right). \quad (20)$$

Here $\Theta$ denotes the learnable components in the dynamics embedding, *e.g.*, initialization $q_\theta^0$, the stepsize $(\eta)$ in the HMC/Langevin updates, and the adaptive part $(S_v, S_x, g_v, g_x)$ in the generalized HMC. The parametrization of the initial distribution is discussed in Appendix C. Compared to the optimization in GANs (Goodfellow et al., 2014; Arjovsky et al., 2017; Dai et al., 2017), beside the reversal of $\min$-$\max$ in (20), the major difference is that our "generator" (the dual sampler) shares parameters with the "discriminator" (the primal potential function). In our formulation, the updates of the potential function automatically push the generator toward the target distribution, thus accelerating learning efficiency. Meanwhile, the tunable parameters in the dynamics embedding are learned adversarially, further promoting the efficiency of the dual sampler. These benefits will be empirically demonstrated in Section 5.

Similar optimization can be derived for generalized HMC (13) with density (17). For the $T$-step stochastic Langevin dynamics embedding (14), we apply the density value (19) to (9), which also leads to a $\max$-$\min$ optimization with multiple momenta.

We use stochastic gradient descent to estimate $f$ for the exponential families as well as the parameters of the dynamics embedding $\Theta$ adversarially. Note that since the generated sample $(x_f^T, v_f^T)$ depends on $f$, the gradient w.r.t. $f$ should also take these variables into account as back-propagation through time (BPTT), *i.e.*,

$$\nabla_f \ell(f; \Theta) = \widehat{\mathbb{E}}_{\mathcal{D}} \left[ \nabla_f f(x) \right] - \mathbb{E}_{q^0} \left[ \nabla_f f\left(x^T\right) \right]$$
$$- \mathbb{E}_{q^0} \left[ \nabla_x f\left(x^T\right) \nabla_f x^T + \lambda v^T \nabla_f v^T \right]. \quad (21)$$

We illustrate the MLE via HMC adversarial dynamics embedding in Algorithm 1. The same technique can be applied to alternative dynamics embeddings parametrized dual sampler as in Appendix B. Considering the dynamics embedding as an *adaptive* sampler that automatically learns w.r.t. different models and datasets, the updates for $\Theta$ can be understood as *learning to sample*.

---

**Algorithm 1** MLE via Adversarial Dynamics Embedding (ADE)

1: Initialize $\Theta_1$ randomly, set length of steps $T$.
2: **for** iteration $k = 1, \ldots, K$ **do**
3:     Sample mini-batch $\{x_i\}_{i=1}^m$ from dataset $\mathcal{D}$ and $\left\{x_i^0, v_i^0\right\}_{i=1}^m$ from $q_\theta^0(x,v)$.
4:     **for** iteration $t = 1, \ldots, T$ **do**
5:         Compute $(x^t, v^t) = \mathbf{L}\left(x^{t-1}, v^{t-1}\right)$ for each pair of $\left\{x_i^0, v_i^0\right\}_{i=1}^m$.
6:     **end for**
7:     **[Learning the sampler]** $\Theta_{k+1} = \Theta_k - \gamma_k \hat{\nabla}_\Theta \ell(f_k; \Theta_k)$
8:     **[Estimating the exponential family]** $f_{k+1} = f_k + \gamma_k \hat{\nabla}_f \ell(f_k; \Theta_k)$.
9: **end for**

---

# 4 Related Work

**Connections to other estimators** The primal-dual view of the MLE also allows us to establish connections between the proposed estimator, *adversarial dynamics embedding* (ADE), and existing approaches, including contrastive divergence (Hinton, 2002), pseudo-likelihood (PL) (Besag, 1975), conditional composite likelihood (CL) (Lindsay, 1988), score matching (SM) (Hyvärinen, 2005), minimum (diffusion) Stein kernel discrepancy estimator (DSKD) (Barp et al., 2019), non-local contrastive objectives (NLCO) (Vickrey et al., 2010), minimum probability flow (MPF) (Sohl-Dickstein et al., 2011), and noise-contrastive estimation (NCE) (Gutmann and Hyvärinen, 2010). As summarized

Table 1: (Fix) dual samplers used in alternative estimators. We denote $p_{\mathcal{D}}$ as the empirical data distribution, $x_{-i}$ as $x$ without $i$-th coordinate, $p_n$ as the prefixed noise distribution, $\mathcal{T}_f(x'|x)$ as the HMC/Langevin transition kernel, $T_{\mathcal{D},f}(x)$ as the Stein variational gradient descent, and $A(x, x')$ as the acceptance ratio.

| Estimators | Dual Sampler $q(x)$ |
|---|---|
| CD | $\int \prod_{i=1}^T \mathcal{T}_f\left(x^i|x^{i-1}\right) A(x^i, x^{i-1}) p_{\mathcal{D}}(x_0) dx_0^{T-1}$ |
| SM | $\int \mathcal{T}_f(x'|x) p_{\mathcal{D}}(x) dx$ |
| DSKD | $x' = T_{\mathcal{D},f}(x)$ |
| PL | $q(x) = \frac{1}{d} \sum_{i=1}^d p_f(x_i|x_{-i}) p_{\mathcal{D}}(x_{-i})$ |
| CL | $q(x) = \frac{1}{m} \sum_{i=1}^m p_f(x_{A_i}|x_{-A_i}) p_{\mathcal{D}}(x_{-A_i})$ $\{A_i\}_{i=1}^m = d$ and $A_i \cap A_j = \emptyset$ |
| NLCO | $\sum_{i=1}^m \int p_{(f,i)}(x) p(S_i|x') p_{\mathcal{D}}(x') dx$ $p_{(f,i)}(x) = \frac{\exp(f(x))}{Z_i(f)}, x \in S_i$ |
| MPF | $\int \mathcal{T}_f(x'|x) \exp\left(\frac{1}{2}(f(x') - f(x))\right) p_{\mathcal{D}}(x) dx$ |
| NCE | $\left(\frac{1}{2} p_{\mathcal{D}} + \frac{1}{2} p_n\right) \frac{\exp(f(x))}{\exp(f(x)) + p_n(x)}$ |

in Table 1, these existing estimators can be recast as the special cases of ADE, by replacing the adaptive dual sampler with hand-designed samplers, which can lead to extra error and inferior solutions. Appendix D gives detailed derivations of the connections.

Exploiting deep models for energy-based model estimation has been investigated in Kim and Bengio (2016); Dai et al. (2017); Liu and Wang (2017); Dai et al. (2019). However, the parametrization of the dual sampler should both be flexible and tractable to achieve better performance. Existing work is limited in one aspect or another. Kim and Bengio (2016) parameterized the sampler via a deep directed graphical model, whose approximation ability is restrictive and the entropy is intractable. Dai et al. (2017) proposed algorithms relying either on a heuristic approximation or a lower bound of the entropy, and requiring learning an extra auxiliary component besides the dual sampler. Dai et al. (2019) applied the Fenchel dual representation twice to reformulate the entropy term, but the algorithm requires knowing a proposal distribution with the same support, which is impractical for high-dimensional data. By contrast, ADE achieves both sufficient flexibility and tractability by exploiting the augmented model and a novel parametrization within the primal-dual view.

**Learning to sample**   ADE also shares some similarity with meta learning for sampling (Levy et al., 2018; Feng et al., 2017; Song et al., 2017; Gong et al., 2019), where the sampler is parametrized via a neural network and learned through certain objectives. The most significant difference lies in the ultimate goal: we focus on exponential family *model estimation*, where the learned sampler *assists* with this objective. By contrast, learning to sample techniques target on a sampler for a *fixed* model. This fundamentally distinguishes ADE from methods that only learn samplers. Moreover, ADE exploits an augmented model that yields tractable entropy estimation, which has not been fully investigated in previous literature.

## 5   Experiments

In this section, we test ADE on several synthetic datasets in Section 5.1 and real-world image datasets in Section 5.2. The details of each experiment setting can be found in Appendix F.

### 5.1   Synthetic experiments

We compare ADE with SM, CD, and primal-dual MLE with the normalizing planar flow (Rezende and Mohamed, 2015) sampler (NF) to investigate the claimed benefits. SM, CD and primal-dual with NF can be viewed as special cases of our method, with either a fixed sampler or restricted parametrized $q_\theta$. Thus, this also serves as an ablation study of ADE to verify the significance of its different subcomponents. We keep the model sizes the same in NF and ADE (10 planar layers). Then we perform 5-steps stochastic Langevin steps to obtain the final samples $x^T$ with standard Gaussian noise in each step, and without incurring extra memory cost. For fairness, we conduct CD with 15 steps. This setup is preferable to CD with an extra acceptance-rejection step. We emphasize that, by comparison to SM and CD, ADE learns the sampler and exploits the gradients through the sampler. In comparison to primal-dual with NF, dynamics embedding achieves more flexibility without introducing extra parameters. Complete experiment details are given in Appendix F.1.

In Figure 1, we visualize the learned distribution using both the learned dual sampler and the unnormalized exponential model on several synthetic datasets. Overall, the sampler almost perfectly recovers the distribution, and the learned $f$ captures the landscape of the distribution. We also plot the convergence behavior in Figure 2. We observe that the samples are smoothly converging to the true data distribution. As the learned sampler depends on $f$, this figure also indirectly suggests good convergence behavior for $f$. More results for the learned models can be found in Figure 5 in Appendix G.

A quantitative comparison in terms of the MMD (Gretton et al., 2012) of the samplers is in Table 2. To compute the MMD, for NF and ADE, we use 1,000 samples from their sam-

Table 2: Comparison on synthetic data using maximum mean discrepancy (MMD $\times 1e^{-3}$).

| Dataset | SM | NF | CD-15 | ADE |
|---|---|---|---|---|
| 2spirals | 5.09 | 0.69 | -0.45 | **-0.61** |
| Banana | 8.10 | 0.88 | -0.31 | **-0.99** |
| circles | 4.90 | 0.76 | -0.83 | **-1.13** |
| cos | 10.36 | 0.91 | 7.15 | **-0.55** |
| Cosine | 8.34 | 2.15 | 0.78 | **-1.09** |
| Funnel | 13.07 | **-0.92** | -0.38 | -0.75 |
| swissroll | 19.93 | 1.97 | 0.20 | **-0.36** |
| line | 10.28 | 0.39 | 10.5 | **-1.30** |
| moons | 41.34 | 0.80 | 2.21 | **-1.10** |
| Multiring | 2.01 | 0.30 | -0.38 | **-1.02** |
| pinwheel | 18.41 | 3.01 | **-1.03** | -0.95 |
| Ring | 9.22 | 161.89 | 0.12 | **-0.91** |
| Spiral | 9.48 | 5.96 | -0.41 | **-0.81** |
| Uniform | 5.88 | 0.00 | **-1.17** | -0.94 |

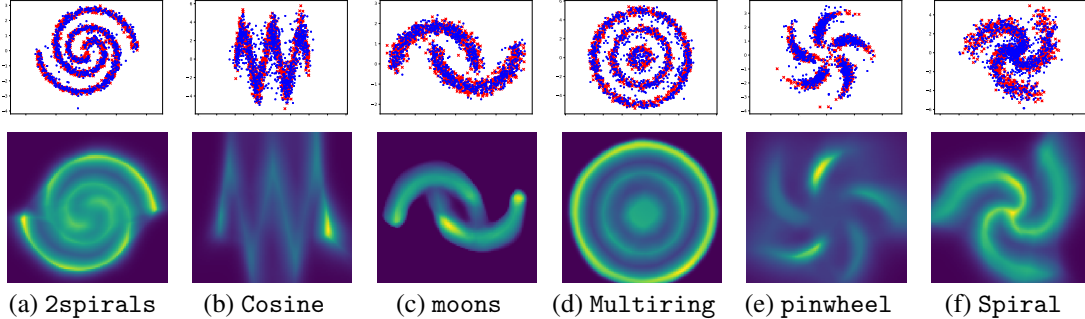

(a) 2spirals    (b) Cosine    (c) moons    (d) Multiring    (e) pinwheel    (f) Spiral

Figure 1: We illustrated the learned samplers from different synthetic datasets in the first row. The × denotes training data and ● denotes the ADE samplers. The learned potential functions $f$ are illustrated in the second row.

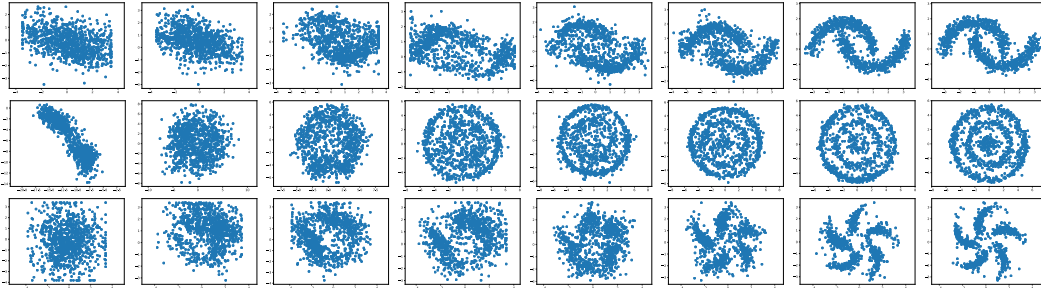

Figure 2: Convergence behavior of sampler on moons, Multiring, pinwheel synthetic datasets.

pler with Gaussian kernel. The kernel bandwidth is chosen using median trick (Dai et al., 2016). For SM, since there is no such sampler available, we use vanilla HMC to get samples from the learned model $f$, and use them to estimate MMD as in Dai et al. (2019). As we can see from Table 2, ADE obtains the best MMD in most cases, which demonstrates the flexibility of dynamics embedding compared to normalizing flow, and the effectiveness of adversarial training compared to SM and CD.

We also investigate the parameters recovery of ADE on the multivariate Gaussians with different dimensions where we know the potential functions. The empirical results can be found in Table 5 in Appendix G. In this simple task, the SM is proven to be consistent and achieve the same estimator as MLE (Hyvärinen, 2005). The objective of ADE can be non-convex due to the learning of the sampler parametrization, therefore, it losses the theoretical guarantees and incurs extra cost. However, as we can see the ADE still achieves comparable performances.

### 5.2 Real-world Image Datasets

We apply ADE to MNIST and CIFAR-10 data. In both cases, we use a CNN architecture for the discriminator, following Miyato et al. (2018), with spectral normalization added to the discriminator layers. In particular, for the discriminator in the CIFAR-10 experiments, we replace all downsampling operations by average pooling, as in Du and Mordatch (2019). We parametrize the initial distribution $p_0(x, v)$ with a deep Gaussian latent variable model (Deep LVM), specified in Appendix C. The output sample is clipped to $[0, 1]$ after each HMC step and the Deep LVM initialization. The detailed architectures and experimental configurations are described in Appendix F.2.

Table 3: Inception scores of different models on CIFAR-10 (unconditional).

| Model | Inception Score |
|---|---|
| WGAN-GP (Gulrajani et al., 2017) | 6.50 |
| Spectral GAN (Miyato et al., 2018) | 7.42 |
| Langevin PCD (Du and Mordatch, 2019) | 6.02 |
| Langevin PCD (10 ensemble) (Du and Mordatch, 2019) | 6.78 |
| ADE: Deep LVM init w/o HMC | 7.26 |
| ADE: Deep LVM init w/ HMC | **7.55** |

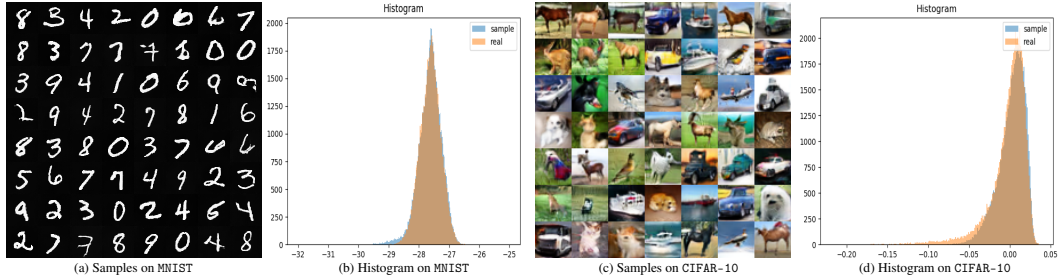

| (a) Samples on MNIST | (b) Histogram on MNIST | (c) Samples on CIFAR-10 | (d) Histogram on CIFAR-10 |

Figure 3: The generated images on MNIST and CIFAR-10 and the comparison between energies of generated samples and real images. The blue histogram illustrates the distribution of $f(x)$ on generated samples, and the orange histogram is generated by $f(x)$ on testing samples. As we can see, the learned potential function $f(x)$ matches the empirical dataset well.

We report the inception scores in Table 3. For ADE, we train with Deep LVM as the initial $q_\theta^0$ with/without HMC steps for an ablation study. The HMC embedding greatly improves the performance of the samples generated by the initial $q_\theta^0$ alone. The proposed ADE not only achieves better performance, compared to the fixed Langevin PCD for energy-based models reported in (Du and Mordatch, 2019), but also enables the generator to outperform the Spectral GAN.

We show some of the generated images in Figure 3(a) and (c); additional sampled images can be found in Figure 6 and 7 in Appendix G. We also plot the potential distribution (unnormalized) of the generated samples and that of the real images for MNIST and CIFAR-10 (using 1000 data points for each) in Figure 3(b) and (d). The energy distributions of both the generated and real images show significant overlap, demonstrating that the obtained energy functions have successfully learned the desired distributions.

Since ADE learns an energy-based model, the learned model and sampler can also be used for image completion. To further illustrate the versatility of ADE, we provide several image completions on MNIST in Figure 4. Specifically, we estimate the model with ADE on fully observed images. For the input images, we mask the lower half with uniform noise. To complete the corrupted images, we perform the learned dual sampler steps to update the lower half of images with

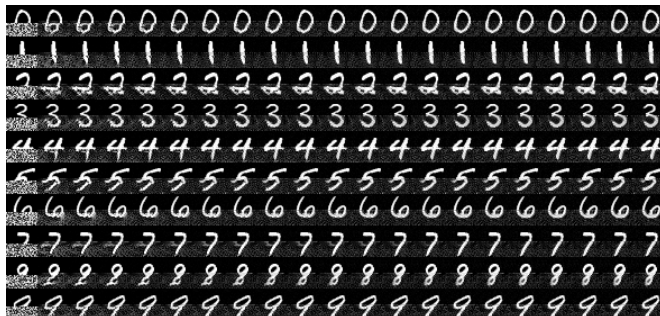

Figure 4: Image completion with the ADE learned model and sampler on MNIST.

the upper half images fixed. We visualize the output from each of the 20 HMC runs in Figure 4. Further details are given in Appendix F.2.

## 6 Conclusion

We proposed Adversarial Dynamics Embedding (ADE) to efficiently perform MLE with general exponential families. In particular, by utilizing the primal-dual formulation of the MLE for an augmented distribution with auxiliary kinetic variables, we incorporate the parametrization of the dual sampler into the estimation process in a fully differentiable way. This approach allows for shared parameters between the primal and dual, achieving better estimation quality and inference efficiency. We also established the connection between ADE and existing estimators. Our empirical results on both synthetic and real data illustrate the advantages of the proposed approach.

### Acknowledgments

We thank David Duvenaud, Arnaud Doucet, George Tucker and the Google Brain team, as well as the anonymous reviewers for their insightful comments and suggestions. L.S. was supported in part by NSF grants CDS&E-1900017 D3SC, CCF-1836936 FMitF, IIS-1841351, CAREER IIS-1350983.

## Footnotes

*indicates equal contribution. Email: {bodai, hadai}@google.com, zhen.liu.2@umontreal.ca.

[1]The code repository is available at https://github.com/lzzcd001/ade-code.

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
