[Supplementary Material · appendix.pdf]

# Appendix

## A Proof of Theorems in Section 3

**Theorem 2 (Equivalent MLE)** *The MLE of the augmented model is the same as the original MLE.*

**Proof** The conclusion is straightforward from independence between $x$ and $v$. We rewrite the MLE (7) in another way as follows

$$\max_f L(f) = \widehat{\mathbb{E}}_{x \sim \mathcal{D}} \left[ \log \int p(x, v) \, dv \right] \tag{22}$$

$$= \widehat{\mathbb{E}}_{x \sim \mathcal{D}} \left[ \log \left( p(x) \int p(v) \, dv \right) \right] \tag{23}$$

$$= \widehat{\mathbb{E}}_{x \sim \mathcal{D}} \left[ \log p(x) + \underbrace{\log \int p(v) \, dv}_{\log 1 = 0} \right] = \widehat{\mathbb{E}}_{x \sim \mathcal{D}} \left[ \log p(x) \right], \tag{24}$$

where the second equation comes from the definition of the $p(x, v)$ in (6) with independent $x$ and $v$. ∎

**Theorem 3 (HMC embeddings as gradient flow)** *In continuous time,* i.e. *with infinitesimal stepsize $\eta \to 0$, the density of particles $(x^t, v^t)$, denoted $q^t(x, v)$, follows the Fokker-Planck equation*

$$\frac{\partial q^t(x, v)}{\partial t} = \nabla \cdot (q^t(x, v) G \nabla \mathcal{H}(x, v)), \tag{25}$$

*with $G = \begin{bmatrix} 0 & \mathbf{I} \\ -\mathbf{I} & 0 \end{bmatrix}$, which has a stationary distribution $p(x, v) \propto \exp(-\mathcal{H}(x, v))$ with the marginal distribution $p(x) \propto \exp(f(x))$.*

**Proof** The first part of the theorem is trivial. When $\eta \to 0$, the HMC follows the dynamical system

$$\left[ \frac{dx}{dt}, \frac{dv}{dt} \right] = [\partial_v \mathcal{H}(x, v), -\partial_x \mathcal{H}(x, v)] = G \nabla \mathcal{H}(x, v).$$

By applying the Fokker-Planck equation, we obtain

$$\frac{\partial q^t(x, v)}{\partial t} = \nabla \cdot (q^t(x, v) G \nabla \mathcal{H}(x, v)). \tag{26}$$

To show that the stationary distribution of such dynamical system converges to $p(x, v) \propto \exp(-\mathcal{H}(x, v))$, recall the fact that

$$\nabla \cdot (G \nabla q^t(x, v)) = -\partial_x \partial_v q^t(x, v) + \partial_v \partial_x q^t(x, v) = 0. \tag{27}$$

The Fokker-Planck equation can be rewritten as

$$\frac{\partial q^t(x, v)}{\partial t} = \nabla \cdot (q^t(x, v) G \nabla \mathcal{H}(x, v) + G \nabla q^t(x, v)). \tag{28}$$

Substitute $p(x, v) \propto \exp(-\mathcal{H}(x, v))$ into (28) and notice

$$\exp(-\mathcal{H}(x, v)) \nabla \mathcal{H}(x, v) + \nabla \exp(-\mathcal{H}(x, v)) = 0,$$

we have $\partial p(x, v) = 0$, which means $p(x, v) \propto \exp(-\mathcal{H}(x, v))$ is a stationary distribution, and thus $p(x) \propto \exp(f(x))$. ∎

**Theorem 4 (Density value evaluation)** *If $(x^0, v^0) \sim q_\theta^0(x, v)$, after $T$ vanilla HMC steps (10), we have*

$$q^T(x^T, v^T) = q_\theta^0(x^0, v^0).$$

*For the $(x^T, v^T)$ from the generalized leapfrog steps (13), we have*

$$q^T(x^T, v^T) = q_\theta^0(x^0, v^0) \prod_{t=1}^{T} (\Delta_x(x^t) \Delta_v(v^t)),$$

where $\Delta_x\left(x^t\right)$ and $\Delta_v\left(v^t\right)$ are defined in (29).

For the $\left(x^T, \{v^i\}_{i=1}^T\right)$ from the stochastic Langevin dynamics (14) with $\left(x^0, \{\xi^i\}_{i=0}^{T-1}\right) \sim q_\theta^0\left(x, \xi\right) \prod_{i=1}^{T-1} q_{\theta_i}\left(\xi^i\right)$, we have

$$q^T\left(x^T, \{v^i\}_{i=1}^T\right) = q_\theta^0\left(x^0, \xi^0\right) \prod_{i=1}^{T-1} q_{\theta_i}\left(\xi^i\right).$$

**Proof** The claim can be obtained by simply applying the change-of-variable rule, *i.e.*,

$$q^T\left(x^T, v^T\right) = q_\theta^0\left(x^0, v^0\right) \prod_{t=1}^T \left|\det \nabla \mathbf{L}_{f,M}\left(x^t, v^t\right)\right|.$$

The Jacobian of the transformation from $(x, v)$ to $\left(x, v^{-\frac{1}{2}}\right)$ is $\begin{bmatrix} \mathbf{I} & \mathbf{0} \\ \frac{\eta}{2}\nabla_x^2 f\left(x\right) & \mathbf{I} \end{bmatrix}$, whose determinant is 1. Similarly, the determinant of the Jacobian of the transform from $\left(x, v^{-\frac{1}{2}}\right)$ to $(x', v')$ is also 1. Therefore, $\left|\det\left(\nabla \mathbf{L}_{f,M}\left(x^t, v^t\right)\right)\right| = 1, \forall i = 1, \ldots, T$, and we prove the first claim.

The second claim can also be obtained in a similar way. By simple algebraic manipulations, we have that the Jacobians of the transformation are all diagonal matrices. Thus,

$$\begin{aligned} \Delta_x\left(x^t\right) &= \left|\det\left(\operatorname{diag}\left(\exp\left(2S_v\left(\nabla_x f\left(x^t\right), x^t\right)\right)\right)\right)\right|, \\ \Delta_v\left(v^t\right) &= \left|\det\left(\operatorname{diag}\left(\exp\left(S_x\left(v^{\frac{1}{2}}\right)\right)\right)\right)\right|. \end{aligned} \quad (29)$$

Similarly, we calculate the Jacobian for the stochastic Langevin update. Specifically, during the $t$-th step, the Jacobian of the transformation from $\left(x^{t-1}, \{v^i\}_{i=1}^{t-1}, \xi^{t-1}\right)$ to $\left(x^{t-1}, \{v^i\}_{i=1}^{t-1}, v^t\right)$ is $\begin{bmatrix} \mathbf{I} & \mathbf{0} & \mathbf{0} \\ \mathbf{0} & \mathbf{I} & \mathbf{0} \\ \frac{\eta}{2}\nabla_x^2 f\left(x\right) & \mathbf{0} & \mathbf{I} \end{bmatrix}$, whose determinant is 1. Similarly, the Jacobian of the transformation from $\left(x^{t-1}, \{v^i\}_{i=1}^{t-1}, v^t\right)$ to $\left(x^t, \{v^i\}_{i=1}^{t-1}, v^t\right)$ is $\begin{bmatrix} \mathbf{I} & \mathbf{0} & \mathbf{0} \\ \mathbf{0} & \mathbf{I} & \mathbf{0} \\ \mathbf{0} & \mathbf{0} & \mathbf{I} \end{bmatrix}$, whose determinant is also 1. Therefore $\left|\det\left(\nabla \mathbf{L}_f\left(x^t, \{v^i\}_{i=1}^t\right)\right)\right| = 1$, which implies

$$q^t\left(x^t, \{v^i\}_{i=1}^{t-1}, v^t\right) = q^{t-1}\left(x^{t-1}, \{v^i\}_{i=1}^{t-1}, \xi^{t-1}\right) = q^{t-1}\left(x^{t-1}, \{v^i\}_{i=1}^{t-1}\right) q_{\theta^{t-1}}\left(\xi^{t-1}\right).$$

Apply the same argument for $\forall t = 1, \ldots, T$, we obtain the third claim. ∎

# B  Variants of Dynamics Embedding

Besides the vanilla Hamiltonian/Langevin embedding and its generalized version we introduced in the main text, we can also embed alternative dynamics, *i.e.*, deterministic Langevin dynamics and its continuous and generalized version.

## B.1  Deterministic Langevin Embedding

We embed the *deterministic Langevin dynamics* to form $x' = \mathbf{L}_{f,M}\left(x\right)$ as $x' = x + \eta\nabla_x f\left(x\right)$ with $x^0 \sim q_\theta^0\left(x\right)$. By the change-of-variable rule, we have $q_{f,M}^T\left(x^T\right) = q_\theta^0\left(x_0\right) \prod_{t=1}^T \left|\det \frac{\partial x^t}{\partial x^{t-1}}\right|$. The deterministic Langevin embedding has been exploited in variational auto-encoder (Dai et al., 2018), in which the variational technique has been applied to bypass the calculation of $\prod_{t=1}^T \left|\det \frac{\partial x^t}{\partial x^{t-1}}\right|$.

Plug such parametrization of the dual distribution into (5), we achieve the alternative objective

$$\max_{f \in \mathcal{F}} \min_{\theta, M, \eta} \ell\left(f; \theta, M, \eta\right) := \widehat{\mathbb{E}}_{\mathcal{D}}\left[f\right] - \mathbb{E}_{x^0 \sim q_\theta^0(x)}\left[f\left(x^T\right) - \log q_\theta^0\left(x\right) - \sum_{t=1}^{T} \log\left|\det \frac{\partial x^t}{\partial x^{t-1}}\right|\right].$$

(30)

For the log-determinant term, $\log\left|\det \frac{\partial x^t}{\partial x^{t-1}}\right| = \log\left|\det\left(I + \eta \mathbf{H}^f\left(x^t\right)\right)\right|$, where $\mathbf{H}_{i,j}^f = \frac{\partial^2 f(x)}{\partial x_i \partial x_j}$. Then, the gradient $\frac{\partial \log\left|\det\left(I + \eta \mathbf{H}^f\left(x^t\right)\right)\right|}{\partial f} = \eta \operatorname{tr}\left(\left(I + \eta \mathbf{H}^f\left(x_t\right)\right)^{-1} \frac{\partial H^f\left(x^t\right)}{\partial f}\right)$. However, the computation of the log-determinant and its derivative w.r.t. $f$ are expensive. We can apply the polynomial expansion to approximate it.

Denoting $\delta$ as the bound of the spectrum of $\mathbf{H}^f\left(x^t\right)$ and $C := \frac{\eta \delta}{1 + \eta \delta}I - \frac{1}{1 + \eta \delta}\mathbf{H}^f\left(x^t\right)$, we have $\lambda\left(C\right) \in \left(-1, 1\right)$. Then,

$$\log\left|\det\left(I + \eta \mathbf{H}^f\left(x^t\right)\right)\right| = d\log\left(1 + \eta \delta\right) + \operatorname{tr}\left(\log\left(I - C\right)\right).$$

We can apply Taylor expansion or Chebyshev expansion to approximate the $\operatorname{tr}\left(\log\left(I - C\right)\right)$. Specifically, we have

- Stochastic Taylor Expansion (Boutsidis et al., 2017) Recall $\log\left(1 - x\right) = -\sum_{k=1}^{\infty} \frac{x^k}{k}$, we have the Taylor expansion

$$\operatorname{tr}\left(\log\left(I - C\right)\right) = -\sum_{i=1}^{k} \frac{\operatorname{tr}\left(C^i\right)}{i}.$$

To avoid the matrix-matrix multiplication, we further approximate the $\operatorname{tr}\left(C\right) = \mathbb{E}_z\left[z^\top C z\right]$ with $z$ as Rademacher random variables, *i.e.*, Bernoulli distribution with $p = \frac{1}{2}$.

Particularly, if we set $i = 1$, recall the $\operatorname{tr}\left(\mathbf{H}^f\left(x\right)\right) = \nabla_x^2 f\left(x\right)$, we can directly calculate without the Hutchinson approximation.

- Stochastic Chebyshev Expansion (Han et al., 2015) We can approximate with Chebyshev polynomial, *i.e.*,

$$\operatorname{tr}\left(\log\left(I - C\right)\right) = \sum_{i=1}^{k} c_i \operatorname{tr}\left(R_i\left(C\right)\right),$$

where $R\left(\cdot\right)$ denotes the Chebshev polynomial as $R_i\left(x\right) = 2x R_{i-1}\left(x\right) - R_{i-2}\left(x\right)$ with $R_1\left(x\right) = x$ and $R_0\left(x\right) = 1$. The $c_i = \frac{2}{k+1}\sum_{j=0}^{k}\log\left(1 - s_j\right)R_i\left(s_j\right)$ if $i \geqslant 1$, otherwise $c_0 = \frac{1}{n+1}\sum_{j=0}^{k}\log\left(1 - s_j\right)$ where $s_j = \cos\left(\frac{\pi\left(k + \frac{1}{2}\right)}{k+1}\right)$ for $j = 0, 1, \ldots, k$.

Similarly, we can use the Hutchinson approximation to avoid matrix-matrix multiplication.

## B.2  Continuous-time Langevin Embedding

We discuss several discretized dynamics embedding above. In this section, we take the continuous-time limit $\eta \to 0$ in the deterministic Langevin dynamics, *i.e.*, $\frac{dx}{dt} = \nabla_x f\left(x\right)$. Follow the change-of-variable rule, we obtain

$$q\left(x'\right) = p\left(x\right)\det\left(I + \eta \mathbf{H}^f\left(x\right)\right)$$
$$\Rightarrow \quad \log q\left(x'\right) - \log p\left(x\right) = -\operatorname{tr}\log\left(I + \eta \mathbf{H}^f\left(x\right)\right) = -\eta \nabla_x^2 f\left(x\right) + \mathcal{O}\left(\eta^2\right).$$

As $\eta \to 0$, we have

$$\frac{d\log q\left(x, t\right)}{dt} = -\nabla_x^2 f\left(x\right). \tag{31}$$

**Remark (connections to Fokker-Planck equation)** Consider the $\frac{dx}{dt} = \nabla_x f\left(x\right)$ as a SDE with zero diffusion term, by Fokker-Planck equation, we obtain the PDE w.r.t. $q\left(x, t\right)$ as

$$\frac{\partial q\left(x, t\right)}{\partial t} = -\nabla \cdot \left(\nabla_x f\left(x\right) q\left(x, t\right)\right).$$

Alternatively, we can also derive the (31) from the Fokker-Planck equation by explicitly writing the derivative. Specifically,

$$
\begin{aligned}
\frac{dq(x,t)}{dt} &= \frac{\partial q(x,t)}{\partial x}\frac{\partial x}{\partial t} + \frac{\partial q(x,t)}{\partial t} \\
&= \frac{\partial q(x,t)}{\partial x}\nabla_x f(x) - \nabla \cdot (\nabla_x f(x) q(x,t)) \\
&= \frac{\partial q(x,t)}{\partial x}\nabla_x f(x) - \nabla_x^2 f(x) q(x,t) - \nabla_x f(x)\frac{\partial q(x,t)}{\partial t} \\
&= -\nabla_x^2 f(x) q(x,t).
\end{aligned}
$$

Therefore, we have

$$
\frac{1}{q(x,t)}\frac{dq(x,t)}{dt} = -\nabla_x^2 f(x) \Rightarrow \begin{bmatrix} \frac{d\log q(x,t)}{dt} = -\nabla_x^2 f(x) \\ \frac{dx}{dt} = \nabla_x f(x) \end{bmatrix}. \tag{32}
$$

Based on (31), we can obtain the samples and its density value by

$$
\begin{bmatrix} x^t \\ \log q(x^t) - \log p_\theta^0(x^0) \end{bmatrix} = \int_{t_0}^{t_1} \begin{bmatrix} \nabla_x f(x(t)) \\ -\nabla_x^2 f(x(t)) \end{bmatrix} dt := \mathbf{L}_{f,t_0,t_1}(x). \tag{33}
$$

We emphasize that this dynamics is different from the continuous-time flow proposed in Grathwohl et al. (2019), where we have $\nabla_x^2 f(x)$ in the ODE rather than a trace operator, which requires one more Hutchinson stochastic approximation. We noticed that Zhang et al. (2018) also exploits the Monge-Ampère equation to design the flow-based model for unsupervised learning. However, their learning algorithm is totally different from ours. They use the parameterization as a new flow and fit the model by matching a *separate* distribution; while in our case, the exponential family and flow share the same parameters and match each other automatically.

We can approximate the integral using a numerical quadrature methods. One can approximate the $\nabla_{(f,t_0,t_1)}\ell(f;t_0,t_1)$ by the derivative through the numerical quadrature. Alternatively, we denote $g(t) = -\frac{\partial\ell(f,t_0,t_1)}{\partial x(t)}$, by the adjoint method, the $\frac{\ell(f,t_0,t_1)}{\partial f}$ is also characterized by ODE

$$
\frac{\partial\ell(f,t_0,t_1)}{\partial f} = \int_{t_0}^{t_1} -g(t)^\top \nabla_f \cdot \nabla_x f(x)\, dt, \tag{34}
$$

and can be approximated by numerical quadrature too.

We can combine the discretized and continuous-time Langevin dynamics by simply stacking several layers of $\mathbf{L}_{f,t_0,t_1}$.

### B.3    Generalized Continuous-time Langevin Embedding

We generalize the continuous-time Langevin dynamics by introducing more learnable space as

$$
\frac{dx}{dt} = h(\xi_f(x)), \tag{35}
$$

where $h$ can be arbitrary smooth function and $\xi_f(x) = (\nabla_x f(x), f(x), x)$. We now derive the distributions formed by such flows following the change-of-variable rule, *i.e.*,

$$
q(x') = p(x)\det(I + \eta\nabla_x h(\xi_f(x)))
$$
$$
\Rightarrow \quad \log q(x') - \log p(x) = -\operatorname{tr}\log(I + \eta\nabla_x h(\xi_f(x))) = -\eta\operatorname{tr}(\nabla_x h(\xi_f(x))) + \mathcal{O}(\eta^2).
$$

As $\eta \to 0$, we have

$$
\frac{d\log q(x,t)}{dt} = -\operatorname{tr}(\nabla_x h(\xi_f(x))). \tag{36}
$$

Similarly, we can compute the samples and its density value by

$$
\begin{bmatrix} x^t \\ \log q(x^t) - \log p_\theta^0(x^0) \end{bmatrix} = \int_{t_0}^{t_1} \begin{bmatrix} h(\xi_f(x)) \\ -\operatorname{tr}(\nabla_x h(\xi_f(x))) \end{bmatrix} dt := \mathbf{L}_{f,t_0,t_1}(x). \tag{37}
$$

# C Practical Algorithm

In this section, we discuss several key components in the implementation of the Algorithm 1, including the gradient computation and the parametrization of the initialization $q_\theta(x, v)$.

## C.1 Gradient Estimator

The gradient w.r.t. $f$ is illustrated in (21). The computation of the gradient needs to compute back-propagated through time, therefore, the computational cost is proportional to the number of sampling steps $T$.

By Denskin's theorem (Bertsekas, 1995), if the samples $(x, v)$ from the optimal solution $p(x, v) \propto \exp(-\mathcal{H}(x, v))$, the third term in (21) exactly vanish to zero, *i.e.*,

$$\nabla_f \ell(f; \Theta) = \widehat{\mathbb{E}}_\mathcal{D} [\nabla_f f(x)] - \mathbb{E}_{(x,v) \sim p(x,v)} [\nabla_f f(x)], \qquad (38)$$

whose computational cost is independent to $T$.

Recall Theorem 3 that as $\eta \to 0$ and $T \to \infty$, the HMC embedding converges to the optimal solution. Therefore, we can approximate the BPTT estimator (21) with the truncated gradient (38). As $T$ increasing, the corresponding dual sampler approaches the optimal solution, and the truncation bias becomes smaller.

## C.2 Initialization Distribution Parametrization

In our algorithm, the dual distribution are parametrized via dynamics sampling method with an initial distribution $q_\theta^0(x, v)$, whose density value is available. There are several possible parametrization:

- **Flow-based model:** The most straightforward parametrization for $q_\theta^0(x, v)$ is utilizing flow-based model (Rezende and Mohamed, 2015; Dinh et al., 2017; Kingma and Dhariwal, 2018). For simplicity, we can decompose $q_\theta^0(x, v) = q_{\theta_1}^0(x) q_{\theta_2}^0(v)$ and parametrized both $q_{\theta_1}^0(x)$ and $q_{\theta_2}^0(v)$ separately.

- **Variants of deterministic Langevin embedding:** The expression ability of flow-based models is still restricted. We can exploit the deterministic Langevin embedding with separate potential function as the initialization. Specifically, we can also decompose $q_\theta^0(x, v) = q_{\theta_1}^0(x) q_{\theta_2}^0(v)$, for the sampler $x$, we exploit

$$x^{t+1} = x^t + \epsilon \phi^t(x^t).$$

Although we do not have the explicit $\log q_{\theta_1}^0(x)$, we can approximate it via either Taylor expansion or Chebyshev expansion as Section B.1. It should be emphasized that in such parametrization, in each layer we use different $\phi^t$ for $t = \{1, \ldots, T\}$, which are all different from $\nabla_x f(x)$.

- **Deep latent variable model:** We can also consider the model

$$v \quad \sim \quad q_{\theta_2}^0(v), \qquad (39)$$
$$x \quad = \quad \phi_{\theta_1}(v) + \epsilon, \quad \epsilon \sim \mathcal{N}(0, \Sigma), \qquad (40)$$

where $q_{\theta_2}^0(v)$ is some known distribution with $\theta_2$ as parameter and $\phi_{\theta_1}$ denotes the neural network with $\theta_1$ as parameter. Therefore, we have the distribution as

$$q_\theta^0(x, v) = \mathcal{N}(x; \phi_{\theta_1}^0(v), \Sigma) q_{\theta_2}^0(v).$$

For vanilla HMC with leap-frog, the auxiliary variable $v$ should be the same size as $x$. However, for generalized HMC, the dimension of $v$ can be smaller than that of $x$.

- **Nonparametric model:** We can also prefix the $q^0(x, v) = q^0(x) q^0(v)$ without learning. Specifically, we set $q^0(x)$ as the empirical $p_\mathcal{D}(x)$ and $q^0(v) = \mathcal{N}(0, \mathbf{I})$. Since the initial distribution is fixed, the learning objective (8) reduces to

$$\max_{f \in q} \min_\Theta \ \ell(f, \Theta) \propto \widehat{\mathbb{E}}_\mathcal{D}[f] - \mathbb{E}_{(x^0, v^0) \sim q^0(x,v)} \left[ f(x^T) - \frac{1}{2} \|v^T\|_2^2 \right]. \qquad (41)$$

# D   Details for Connections to Other Estimators

We provide the details for recasting the existing estimators as special cases of our ADEas listed in Table 1.

## D.1   Connection to Contrastive Divergence

The CD algorithm (Hinton, 2002) is a special case of the proposed algorithm. By Theorem 1, the optimal solution to the inner optimization is $p(x, v) \propto \exp(-\mathcal{H}(x, v))$. Applying Danskin's theorem (Bertsekas, 1995), the gradient of $L(f)$ w.r.t. $f$ is

$$\nabla_f L(f) = \widehat{\mathbb{E}}_{\mathcal{D}}\left[\nabla_f f(x)\right] - \mathbb{E}_{p_f(x)}\left[\nabla_f f(x)\right]. \tag{42}$$

To estimate the integral $\mathbb{E}_{p_f}[\nabla_f f(x)]$, the CD algorithm approximates the negative term in (42) stochastically with a finite MCMC step away from empirical data.

In the proposed dual sampler, by setting $p_\theta^0(x)$ to be the empirical distribution and eliminating the sampling learning, the dynamic embedding will collapse to CD with $T$-HMC steps if we remove gradient through the sampler, *i.e.*, ignoring the third term in (21). Similarly, the persistent CD (PCD) (Tieleman, 2008) and recent ensemble CD (Du and Mordatch, 2019) can also be recast as special cases by setting the negative sampler to be MCMC with initial samples from previous model and ensemble of MCMC samplers, respectively.

From this perspective, the CD and PCD algorithms induce errors not only from the sampler, but also from the gradient back-propagation truncation. The proposed algorithm escapes these sources of bias by learning to sample, and by adopting true gradients, respectively. Therefore, the proposed estimator is expected to achieve better performance than CD as demonstrated in the empirical experiments Section 5.2.

## D.2   Connection to Score Matching

The score matching (Hyvärinen, 2005) estimates the exponential family by minimizing the Fisher divergence, *i.e.*,

$$L_{SM}(f) := -\mathbb{E}_{\mathcal{D}}\left[\sum_{i=1}^{d}\left(\frac{1}{2}(\partial_i f(x))^2\right) + \partial_i^2 f(x)\right]. \tag{43}$$

As explained in Hyvärinen (2007), the objective (43) can be derived as the 2nd-order Taylor approximation of the MLE with 1-step Langevin Monte Carlo as the dual sampler. Specifically, the Langevin Monte Carlo generates samples via

$$x' = x + \frac{\eta}{2}\nabla_x f(x) + \sqrt{\eta}\xi, \quad \xi \sim \mathcal{N}(0, I),$$

then, a simple Taylor expansion gives

$$\log p_f(x') = \log p_f(x) + \sum_{i=1}^{d}\partial_i f(x)\left(\frac{\eta}{2}\partial_i f(x) + \sqrt{\eta}\xi_i\right) + \eta\sum_{i,j=1}^{d}\xi_i\xi_j\partial_{ij}^2 f(x) + o(\eta).$$

Plug such into the negative expectation in $L(f)$, leading to

$$L(f) \approx \widehat{\mathbb{E}}_{\mathcal{D}}\left[\log p_f(x) - \mathbb{E}_{x'|x}[\log p_f(x')]\right] \approx -\eta\mathbb{E}_{\mathcal{D}}\left[\sum_{i=1}^{d}\left(\frac{1}{2}(\partial_i f(x))^2\right) + \partial_i^2 f(x)\right],$$

which is exactly the scaled $L_{SM}(f)$ defined in (43).

Therefore, the score matching can be viewed as applying Taylor expansion approximation with fixed 1-step Langevin sampler in our framework, which is compared in Section 5.1.

## D.3   Connection to Minimum Stein Discrepancy Estimator

The minimum Stein discrepancy estimator (Barp et al., 2019) is obtained by minimizing the Stein discrepancy, including the diffusion kernel Stein discrepancy (DKSD) and diffusion score matching. Without loss of the generality, for simplicity, we recast the DKSD with an identity diffusion matrix as a special approximation to the MLE.

The identity DKSD maximizes the following objective,

$$L_{DKSD}\left(f\right) := - \sup_{h \in \mathcal{H}_k, \|h\|_{\mathcal{H}_k} \leqslant 1} \widehat{\mathbb{E}}_{\mathcal{D}}\left[\mathcal{S}_f h\left(x\right)\right] = -\widehat{\mathbb{E}}_{x, x' \sim \mathcal{D}}\left[\mathcal{S}_f\left(x, \cdot\right) \otimes_k \mathcal{S}_f\left(x', \cdot\right)\right] \qquad (44)$$

where $\mathcal{S}_f h\left(x\right) := \left\langle \mathcal{S}_f\left(x, \cdot\right), h\right\rangle = \left\langle \nabla_x f\left(x\right)^\top k\left(x, \cdot\right) + \nabla k\left(x, \cdot\right), h\right\rangle$.

In fact, the objective (44) can be derived as the Taylor approximation of the MLE with Stein variational gradient descent (SVGD) as the dual sampler. Specifically, the SVGD generates samples via

$$x' = T_{\mathcal{D}, f}\left(x\right) := x + \eta h^*_{\mathcal{D}, f}\left(x\right), \quad x \sim p_{\mathcal{D}}\left(x\right),$$

where $h^*_{\mathcal{D}, f}\left(\cdot\right) \propto \mathbb{E}_{y \sim \mathcal{D}}\left[\mathcal{S}_f\left(y, \cdot\right)\right]$. Then, by Taylor-expansion, we have

$$f\left(x'\right) = f\left(x\right) + \eta \nabla_x f^\top\left(x\right) h^*_{\mathcal{D}, f}\left(x\right) + o\left(\eta\right).$$

We apply the change-of-variable rule, leading to $q\left(x'\right) = p_{\mathcal{D}}\left(x\right) \det\left|\frac{\partial x}{\partial x'}\right|$, therefore,

$$
\begin{aligned}
\log q(x') &= \log p_{\mathcal{D}}\left(x\right) + \log \det\left|\frac{\partial x}{\partial x'}\right| \\
&= \log p_{\mathcal{D}}\left(x\right) - \log \det\left|\frac{\partial x'}{\partial x}\right| \\
&= \log p_{\mathcal{D}}\left(x\right) - \log \det\left|I + \eta \nabla_x h^*_{\mathcal{D}}\left(x\right)\right| \\
&= \log p_{\mathcal{D}}\left(x\right) - \eta \operatorname{tr}\left(\nabla_x h^*_{\mathcal{D}}\left(x\right)\right),
\end{aligned}
$$

where the last equation comes from Taylor expansion.

Plug these into the primal-dual view of MLE (5) with the fixed SVGD dual sampler, we have

$$
\begin{aligned}
L\left(f\right) &\approx \widehat{\mathbb{E}}_{x \sim \mathcal{D}}\left[f\left(x\right) - f\left(x'\right) + \log q\left(x'\right)\right] \\
&= \widehat{\mathbb{E}}_{x \sim \mathcal{D}}\left[-\eta \nabla_x f^\top\left(x\right) h^*_{\mathcal{D}, f}\left(x\right) - \eta \operatorname{tr}\left(\nabla_x h^*_{\mathcal{D}}\left(x\right)\right)\right] + \widehat{\mathbb{E}}_{x \sim \mathcal{D}}\left[\log p_{\mathcal{D}}\left(x\right)\right] + o\left(\eta\right) \\
&= -\eta \underbrace{\widehat{\mathbb{E}}_{x, x' \sim \mathcal{D}}\left[\mathcal{S}_f\left(x, \cdot\right) \otimes_k \mathcal{S}_f\left(x', \cdot\right)\right]}_{L_{DSKD}(f)} + \mathtt{const} + o\left(\eta\right),
\end{aligned}
$$

which is the scaled $L_{DSKD}\left(f\right)$ defined in (44).

Therefore, the (diffusion) Stein kernel estimator can be viewed as Taylor expansion with fixed 1-step Stein variational gradient descent dual sampler in our framework.

### D.4 Connection to Pseudo-Likelihood and Conditional Composite Likelihood

The pseudo-likelihood estimation (Besag, 1975) is a special case of the proposed algorithm by restricting the parametrization of the dual distribution. Specifically, denote the $p_f\left(x_i|x_{-i}\right) = \frac{\exp(f(x_i, x_{-i}))}{Z(x_{-i})}$ with $Z\left(x_{-i}\right) := \int \exp\left(f\left(x_i, x_{-i}\right)\right) dx_i$, instead of directly maximizing likelihood, the pseudo-likelihood estimator is maximizing

$$L_{PL}\left(f\right) := \widehat{\mathbb{E}}_{\mathcal{D}}\left[\sum_{i=1}^{d} \log p_f\left(x_i|x_{-i}\right)\right]. \qquad (45)$$

Then, the $f$ is updated by the following the gradient of $L_{pl}\left(f\right)$, i.e.,

$$\nabla_f L_{PL}\left(f\right) \propto \widehat{\mathbb{E}}_{\mathcal{D}}\left[\nabla_f f\left(x\right)\right] - \mathbb{E}_{i \sim \mathcal{U}(d)}\widehat{\mathbb{E}}_{x_{-i}}\mathbb{E}_{p_f(x_i|x_{-i})}\left[\nabla_f f\left(x_i, x_{-i}\right)\right].$$

The pseudo-likelihood estimator can be recast as a special case of the proposed framework if we fix the dual sampler as **i)**, sample $i \in \{1, \ldots, d\}$ uniformly; **ii)**, sample $x \sim \mathcal{D}$ and mask $x_i$; **iii)**, sample $x_i \sim p_f\left(x_i|x_{-1}\right)$ and compose $\left(x_i, x_{-i}\right)$.

The conditional composite likelihood (Lindsay, 1988) is a generalization of pseudo-likelihood by maximizing

$$L_{CL}\left(f\right) := \widehat{\mathbb{E}}_{\mathcal{D}}\left[\sum_{A_i=1}^{m} \log p_f\left(x_{A_i}|x_{-A_i}\right)\right], \qquad (46)$$

where $\{A_i\}_{i=1}^{m} = d$ and $A_i \cap A_j = \emptyset$. Similarly, the composite likelihood is updating with prefixed conditional block sampler for negative sampling.

Same as CD, the prefixed sampler and the biased gradient in pseudo-likelihood and composite likelihood estimator will induce extra errors and lead to inferior solution. Moreover, the pseudo-likelihood may not applicable to the general exponential family with continuous variables, whose conditional distribution is also intractable.

## D.5 Connection to Non-local Contrastive Objectives

The non-local contrastive estimator (Vickrey et al., 2010) is obtained by maximizing

$$L_{NCO}(f) := \widehat{\mathbb{E}}_{\mathcal{D}}\left[\sum_{i=1}^{m} w(x, S_i)(f(x) - \log Z_i(f))\right], \tag{47}$$

where $[S_i]_{i=1}^{m}$ denotes some prefixed partition of $\Omega$, $Z_i(f) = \int_{x \in S_i} \exp(f(x))\, dx$, and $w(x, S_i) = P(x \in S_i|x)$ with $\sum_{i=1}^{m} w(x, S_i) = 1$. The objective (47) leads to the update direction as

$$\nabla_f L_{NCO}(f) = \widehat{\mathbb{E}}_{\mathcal{D}}[\nabla_f f(x)] - \mathbb{E}_{q_f(x)}[\nabla_f f], \tag{48}$$

where $q_f(x) = \sum_{i=1}^{m} \int p_{(f,i)}(x) w(x', S_i) p_{\mathcal{D}}(x')\, dx'$ with $p_{\mathcal{D}}$ as the empirical distribution and $p_{(f,i)}(x) = \frac{\exp(f(x))}{Z_i(f)}$, $x \in S_i$. Therefore, the non-local contrastive objective is a special case of the proposed framework with the dual sampler as **i)**, sample $x'$ uniformly from $\mathcal{D}$; **ii)**, sample $S_i$ conditional on $x'$ according to $w(x, S_i)$; **iii)**, sample $x_i \sim p_{(f,i)}(x)$ within $S_i$. Such negative sampling method is also not applicable to the general exponential family with continuous variables.

## D.6 Connection to Minimum Probability Flow

In the continuous state model, the minimum probability flow (Sohl-Dickstein et al., 2011) estimates the exponential family by maximizing

$$L_{MPF}(f) := -\widehat{\mathbb{E}}_{x \sim \mathcal{D}}\mathbb{E}_{x' \sim \mathcal{T}_f(x'|x)}\left[\exp\left(\frac{1}{2}(f(x') - f(x))\right)\right],$$

where $\mathcal{T}_f$ is a *hand-designed* symmetric transition kernel based on the potential function $f(x)$, *e.g.*, Hamiltonian or Langevin simulation. Then, the MPF update direction can be rewritten as

$$\widehat{\mathbb{E}}_{x \sim \mathcal{D}}\mathbb{E}_{x' \sim \Gamma(x'|x)}[\nabla_f f(x) - \nabla_f f(x') - \nabla_x f(x')\nabla_f x']. \tag{49}$$

where $\Gamma(x'|x) := \mathcal{T}_f(x'|x)\exp\left(\frac{1}{2}(f(x') - f(x))\right)$. The probability flow operator $\Gamma(x'|x)$ actually defines a Markov chain sampler that achieves the following balance equation,

$$\Gamma(x'|x) p_f(x) = \Gamma(x|x') p_f(x').$$

Similar to CD and score matching, the MPF exploits the 1-step MCMC. Moreover, the gradient in MPF also considers the effects in sampler as the third term in (49). Therefore, the MPF can be recast as a special case of our algorithm with the prefixed dual sampler as $x \sim \mathcal{D}$ and $x' \sim \Gamma(x'|x)$.

## D.7 Connection to Noise-Contrastive Estimator

Instead of directly estimating the $f$ in the exponential family, Gutmann and Hyvärinen (2010) propose the noise-contrastive estimation (NCE) for the density ratio between the exponential family and some user defined reference distribution $p_n(x)$, from which the parameter $f$ can be reconstructed. Specifically, the NCE considers an alternative representation of exponential family distribution as $p_f(x) = \exp(f(x))$, which explicitly enforces $\int \exp(f(x))\, dx = 1$. The NCE is obtained by maximizing

$$L_{NCE}(f) := \widehat{\mathbb{E}}_{\mathcal{D}}[\log h(x)] + \mathbb{E}_{p_n(x)}[\log(1 - h(x))], \tag{50}$$

where $h(x) = \frac{\exp(f(x))}{\exp(f(x)) + p_n(x)}$. Then, we have the gradient of $L_{NCE}(f)$ as

$$\nabla_f L_{NCE}(f) = \widehat{\mathbb{E}}_{\mathcal{D}}[\nabla_f f(x)] - \mathbb{E}_{\frac{1}{2}p_{\mathcal{D}} + \frac{1}{2}p_n}[h(x)\nabla_f f(x)]. \tag{51}$$

The negative sampler in the (51) can be understood as an approximate importance sampling algorithm where the proposal is $\frac{1}{2}p_{\mathcal{D}} + \frac{1}{2}p_n$ and the reweighting part is $h(x)$. As the $\exp(f)$ approaching $p_{\mathcal{D}}$, the $h(x)$ will approach the true ratio $\frac{\exp(f(x))}{p_{\mathcal{D}} + p_n(x)}$, and thus, the negative samples will converge to true model samples.

The NCE can be understood as learning an important sampler. However, the performance of NCE highly relies on the quality $h(x)$, *i.e.*, the choice of $p_n(x)$. It is required to cover the support of $p_\mathcal{D}(x)$, which is non-trivial in practical high-dimensional applications.

# E   More Related Work

The parametrization of the dual sampler should be both flexible enough and density tractable to achieve better performance. Pioneering works are limited in either one aspect or other. Kim and Bengio (2016) parameterize the sampler via a deep directed graphical model, whose approximation ability is limited to known distributions. Meanwhile, they fit $q$ by minimizing the $KL$-divergence with an approximation of the entropy term, leading to unclear relationship to MLE. Due to the difficulty of the entropy term for general transport mapping parametrization, a variety of approximate surrogates have been proposed to relax the density value tractability requirement. Liu and Wang (2017) learn the sampler $q$ to mimic the Stein variational gradient descent sampling procedure without a consistent objective; Dai et al. (2017) propose algorithms relying on either a heuristic approximation or a lower bound of the entropy, with extra auxiliary component introduced to be learned; Dai et al. (2019) apply a second Fenchel dual representation to reformulate the entropy term, at the cost of introducing another auxiliary function to be estimated. Meanwhile, the second Fenchel duality parametrization relies on a proposal distribution with the same support for numerical stability, which is impractical for high-dimensional data. In contrast to these existing methods, the proposed dynamics embedding achieves both flexibility and tractability of entropy estimation with less independent auxiliary parameters introduced.

One of our major contributions is learning a sampling strategy for the exponential family estimation through the primal-dual view of MLE. The proposed algorithm shares some similarities with recent advances in meta learning for sampling (Levy et al., 2018; Feng et al., 2017; Song et al., 2017; Gong et al., 2019), in which the sampler is parametrized via neural network and will be learned through certain objectives. However, we emphasize that the most significant difference lies in the ultimate goals: we focus on exponential family *model estimation* and the learned sampler is only introduced to *assist* with this objective. By contrast, the learning to sample techniques are targeting on learning a *fixed* model that is already given. This fundamentally distinguishes the proposed ADE from methods that only learns samplers, leading to totally different learning criterion and algorithm updates, *i.e.*, the primal model will be learned back through the learned sampler, from which perspective the proposed algorithm can be understood as meta[2]-learning.

# F   Experiment Details

## F.1   Synthetic Experiments Details

We parametrize the potential function $f$ with fully connected multi-layer perceptron with 3 hidden layers. Each hidden layer has 128 hidden units. We use ReLU to do the nonlinear activation in each hidden layer. We clip the norm of $\nabla_x f$ when updating $v$, and clip $v$ when updating $x$. The coefficient $\lambda$ in (20) is tuned in $\{0.1, 0.5, 1\}$. For the NF baseline, we tune the number of layers in $\{10, 15, 20\}$. For our ADE, we fix the number of normalizing flow layers to be 10, and then perform at most 10 steps of dynamics updates. So finally, the number of steps for sampling is comparable, while the ADE maintains less memory cost.

To make the training stable, we also tried several tricks, including:

1. clip samples in HMC. This helps stabilize the training; We assume the final output has limited support over 2D space.

2. gradient penalty for $f(\cdot)$. We use a small penalty coefficient 0.01 for this, which is not very important though.

3. variance of proposal gaussian distribution. While we use 1 in general, a standard deviation of 0.5 would be more helpful in some cases.

4. penalty of momentum term in HMC. This is equivalent to the variance of prior of the latent variable we introduced.

The dataset generators are collect from several open-source projects [2] [3]. During training, we use this generator to generate the data from the true distribution on the fly. To get a quantitative comparison, we also generate 1,000 data samples for held-out evaluation. We illustrate the unnormalized model $\exp(c \cdot f)$ in Figure 1 and 5, where $c$ is a constant that is tuned within $[0.01, 10]$.

To compute the MMD, for NF and ADE, we use 1,000 samples from their sampler with Gaussian kernel. The kernel bandwidth is chosen using median trick (Dai et al., 2016). For SM, since there is no such sampler available, we directly use vanilla HMC to get sample from the learned model $f$, and use them to estimate MMD.

**Parameter estimation experiments** In the experiment of recovering parameters of a given graphical model from data, we use high dimensional gaussian distribution with diagonal covariance. Here the energy function to be estimated $f(x) = -0.5(x - \mu)^\top \Sigma^{-1}(x - \mu)$, where $\Sigma$ is a diagonal matrix.

For our method, we use a 2-layer MLP as initial proposal with 3 of HMC steps afterwards. The step size in HMC is learned end-to-end. For CD, we use up to 15 steps of HMC, where the step size is adaptively adjusted according to the rejection rate. For all the methods, we average the parameters estimated in the last 5 epochs during training, and report the best results in this parameter estimation procedure.

### F.2 Real-world Experiments Details

Table 4: Our architectures for both potential function $f(x)$ and initial dual sampler $p_\theta^0(x, v)$ used in `MNIST` and `CIFAR-10` experiments.

| Potential function $f(\cdot)$ |
| --- |
| 3x3 conv, 64 |
| 3x3 conv, 128 |
| 2x2 avg pool |
| 3x3 conv, 128 |
| 3x3 conv, 256 |
| 2x2 avg pool |
| 3x3 conv, 256 |
| 7x7 avg pool |
| fc, $256 \to 1$ |

(a) Potential function $f(\cdot)$

| Initial dual sampler |
| --- |
| fc, $512 \to 4 \times 4 \times 512$ |
| Reshape to $4 \times 4$ Feature Map |
| 2x2 Deconv, 256, stride 2 |
| 2x2 Deconv, 128, stride 2 |
| 2x2 Deconv, 64, stride 2 |
| 3x3 Deconv, 3, stride 1 |

(b) initial dual sampler

We used the standard spectral normalization on the discriminator to stabilize the training process, and Adam with learning rate $10^{-4}$ and $\beta_1 = 0.0$ to optimize our model. For stability, we use a separate Adam optimizer for the hmc parameters and set the epsilon to $1e - 5$. We trained the models with 200000 iterations with batch size being 64. For better performance, we used generalized HMC (13), where we set $S_v(\cdot) = 0$, $S_x(\cdot) = 0$, $g_v(v) = \text{clip}(v, -0.01, 0.01)$ and $g_x(v^{1/2}) = v^{1/2}$. We fix $\eta$ to be 0.5. The step sizes for our HMC sampler are independently learned for all HMC dimensions but shared among all time steps, and the values are all initialized to 10. We set the number of HMC steps to 30. The coefficient of the entropy regularization term is set to $10^{-5}$ and that of the $L_2$ regularization on the momentum vector in the last HMC step is set to $10^{-5}$.

We demonstrate the architectures of potential function $f$ and initial Deep LVM in Table 4. A leaky ReLU follows each convolutional/deconvolutional layer in both the discriminator and generator. For the discriminator, we use spectral normalization for all layers in the discriminator. In addition, there is no activation function after the final fully-connected layer. For each deconvolution layer in the generator, we insert a batch normalization layer before passing the output to the leaky ReLU.

We generate the image from the model and illustrated in Figure 3, Figure 6 and Figure 7. We also compared in terms of inception score with other energy-model training algorithm and several state-of-the-art GAN algorithm in Table 3, where the ADE achieves the best performances. Also,

with simple importance sampling and proposal distribution being uniform distribution on $[-1, 1]^{n_d}$ ($n_d$ is the dimension of images), the log likelihood (in nats) on `CIFAR-10` is estimated to be around 2100.

We also trained a non-parametric ADE on `MNIST` dataset for image completion to verify our algorithm. Specifically, we use with the same discriminator architecture used in parametric ADE for `MNIST`. The model is trained with fully observed images. We used generalized HMC (13), where we set $S_v(v)$ being a learnable logit (so that $\exp(S_v(\cdot)) \in [0, 1]$), $g_v(v) = \text{clip}(v, -0.1, 0.1)$, $S_x(\cdot) = 0$ and $g_x(\cdot) = 1$. Both $S_v$ and $\eta$ will be learned, with $\eta$ initialized to $\sqrt{10}$ and $S_v$ initialized to a small number close to 0. We unfold 60 steps of HMC in the dual samplers. As in Du and Mordatch (2019), we used a replay buffer of size 10000. We added extra amount of noise into the dataset to make the training process more stable. We trained the model with Adam optimizer ($\beta_1 = 0.0, \beta_2 = 0.999$) for 60000 iterations.

We tested the ADE by image completion where we covered the lower half of images with uniform noise and used them as input to the learned HMC operators. We repeatedly apply the learned HMC with the learned model to lower half of these images for 20 steps, with the upper half images fixed, and obtain $\text{HMC}^{(20)}(x_0; S_v, \eta)$. We visualize the output from each of the 20 HMC runs in Figure 4.

## G   More Experiment Results

**More results on synthetic datasets**   We visualized the learned models and samplers on all the synthetic datasets in Figure 5.

Figure 5: Learned samplers in odd row and potential function $f$ in even row from different synthetic datasets. In the sampler illustration in odd rows, the $\times$ denotes training data and $\bullet$ denotes the ADE samplers.

**More results on parameters recovery**   We have conducted empirical comparion between ADE, CD and SM on multivariate Gaussians with different dimensions, where we know the potential functions, to investigate the effect of the number of dimensionality and complexity of the potential function on these algorithms.

Table 5: Parameter recovering on Multivariate Gaussians.

| Dataset | SM | CD-5 | ADE |
|---|---|---|---|
| 2D-Gaussian | $\mathbf{2.18 \times 10^{-3}}$ | $5.67 \times 10^{-3}$ | $2.28 \times 10^{-3}$ |
| 5D-Gaussian | $3.17 \times 10^{-3}$ | $4.19 \times 10^{-1}$ | $\mathbf{3.09 \times 10^{-3}}$ |
| 10D-Gaussian | $3.90 \times 10^{-3}$ | $6.36 \times 10^{-1}$ | $\mathbf{3.23 \times 10^{-3}}$ |

The 5 runs average results, in terms of RMSE between learned parameters and the true parameters, are reported in Table 5.

**More results on real-world image generation**   We illustrated additional generated images by the proposed ADE on MNIST and CIFAR-10 in Figure 6 and Figure 7, respectively.

Figure 6: Generated images for MNIST by ADE.

Figure 7: Generated images for CIFAR-10 by ADE.

## Footnotes

[2]https://github.com/rtqichen/ffjord.

[3]https://github.com/kevin-w-li/deep-kexpfam.