[Reviews · NeurIPS 2019]

Reviewer 1



Update: I thank the authors for their responses and think they did a good job to address the concerns raised by the reviewers. Therefore, I'll hold my original score and believe the paper makes a strong contribution for estimating exponential family models and should be accepted by NeurIPS. ----------------------------------------------------------------------------------------------------------- Overall: This paper proposes a novel Adversarial Dynamics Embedding (ADE) that directly approximates the MLE of general exponential family distributions while achieving computational and statistical efficiency. It not only provides theoretical justification for the proposed dynamics embeddings, but also show how other estimators can be framed as special case in the proposed framework. Quality: The overall approach proposed by the paper seems to be technically sound. The paper did a good job in comparing the proposed method to other estimators and show how the proposed method overcome the limitations of other estimators. The results on both synthetic and real-data (MNIST and CIFAR-10) show the flexibility of the proposed dynamics embeddings without introducing extra parameters. Clarity: The paper is well-written. Originality: The proposed Adversarial Dynamics Embedding is entirely novel to my knowledge. Significance: This paper makes a good contribution to algorithms/methods for estimating general exponential family distributions. It overcomes the shortcomings of existing estimators, and can directly approximate the MLE while improving computational and statistical efficiency.

Reviewer 2



# Originality This is an original approach to MLE of exponential families based on a representation of the intractable log partition function using Fenchel duality. Fenchel duality introduces a dual sampler that needs to be optimized, resulting in an "adversarial" max-min problem. The authors propose to estimate an augmented model inspired by Hamiltonian Monte Carlo, which allows them to design dual samplers generated by Hamiltonian dynamics in the original space or an embedding space. # Quality The quality of the paper is quite good but could be improved. For example, references need to be fixed in the main paper and the appendix. Tests reporting MLEs for some common exponential families would be useful. # Clarity Most parts of the paper are clearly written. Personally, I think that the fragmented style splitting the text into remarks perturbs the flow of the text. # Significance I think this is a very nice and flexible framework for MLE of exponential families that should result in powerful estimators and provide a starting point for further algorithmic developments. What about exponential families defined on discrete spaces?

Reviewer 3



This paper starts from introducing max-min formulation of MLE by using Fenchel dual of log partition function. Authors focus on providing a better solution of min part (i.e. dual sampler). They first construct augmented model by introducing an auxiliary momentum variable. The essence is to incorporate the model with an independent Gaussian variable, see (6). By this construction, the min problem is equivalent to (8). They further conduct T HMC-based steps, (13), to approximately solve it and show how the output approximates the density value and helps with max problem. They compare the proposed scheme with SM, CD etc to show superiority. The paper is readable and has good quality overall. The experimental results are significant. But I have some brief concerns: 1, the significance of the paper need to be clarified more clearly. In particular, the max-min problem is from plugging the Fenchel dual of log partition, which is standard. For the augmented MLE, since later authors use HMC to represent dual sampler and HMC has natural augmented interpretation, (6) is more likely proposed due to the specific HMC scheme adopted, instead of an original work. 2, the finer analysis of Algorithm 1 is lacking for study. Specifically, is it possible that SGD for max problem would blur the precision of HMC such that the inner iteration number T need to adapt with outer iteration suitably. The theorem only shows HMC is able to well approximate exponential family, which I think is standard, but the combination with SGD for max problem need to discuss as well. 3, for the experiment, instead of using fixed number of steps, using CV to select for all methods is preferred. I don't see why T = 5 for ADE corresponds to T = 15 for CD. Also the setup of leapfrog stepsize is not mentioned in the main paper. 4, minor thing: should it be log(Z(f)) in (7)?

[Author Response · NeurIPS 2019]

We thank the reviewers for their close reading, detailed comments, and overall positive assessment. We address the questions raised by each reviewer separately.

**Reviewer 1:** Thanks for the appreciation and suggestions for paper refinement. We will fix the typos and polish the figures for the final version.

**Reviewer 2:** We will improve the flow and formatting of the paper, and fix the references in the final version.

- **Empirical experiments on known exponential family distributions.** We have conducted a new experiment comparing ADE with CD and SM on multivariate Gaussians with different dimensions and `banana` datasets, where we know the potential functions, to investigate the effect of the number of dimensionality and complexity of the potential function on these algorithms. For fairness, we compare the ADE with

Table 1: Parameter recovering on synthetic datasets.

| Dataset | SM | CD-5 | ADE |
|---|---|---|---|
| 2D-Gaussian | $\mathbf{2.18 \times 10^{-3}}$ | $5.67 \times 10^{-3}$ | $2.28 \times 10^{-3}$ |
| 5D-Gaussian | $3.17 \times 10^{-3}$ | $4.19 \times 10^{-1}$ | $\mathbf{3.09 \times 10^{-3}}$ |
| 10D-Gaussian | $3.90 \times 10^{-3}$ | $6.36 \times 10^{-1}$ | $\mathbf{3.23 \times 10^{-3}}$ |
| Banana | $2.33 \times 10^{-2}$ | $6.72 \times 10^{-2}$ | $\mathbf{6.00 \times 10^{-3}}$ |

HMC-3 on 2-layer MLP to CD with HMC-5, which has the same number of operations. The models are trained with 1000 samples. The 5 runs average results, in terms of RMSE between learned parameters and the true parameters, are reported in Table 1. As we can see, ADE consistently achieves comparable or the best performance. We will add these comparison and a detailed analysis to the final version.

- **ADE for exponential families on discrete variables.** For an enumerable discrete space, the partition function is tractable and the MLE can be computed exactly. For a combinatorial discrete space, our primal-dual MLE is still valid. The major difficulty lies in the dual sampler parametrization, where the HMC/Langevin embeddings are not applicable since the gradient is not well-defined. We are exploring alternative sampling algorithm embeddings, *e.g.*, the importance sampler and Gibbs sampler, for future work.

- **ADE limitations and how to overcome.** The cost of gradient computation in respect of the potential function will be proportional to the number of sampling layers $T$. We provide a gradient approximation which is independent of $T$, however, using Danskin's theorem. See Appendix C for details.

- **Parameter tuning in ADE.** When we use neural networks to parametrize $q^0(x, v)$ and the generalized HMC layers in ADE, then the parameter tuning requirements for ADE and GANs are comparable, *i.e.*, we tune the inner optimization stepsize and the schedule between the inner minimisation and outer maximisation. When we use a nonparametric $q^0(x, v)$ and a vanilla HMC layer, the parameter tuning requirements of ADE and CD are comparable, *i.e.*, we tune the optimization stepsize in ADE, and the leapfrog step size for CD.

**Reviewer 3:**

- **Main contribution.** Re: "[the authors] further conduct T vanilla HMC steps to approximately solve it." We are most certainly *not* using a fixed (vanilla) HMC sampler to directly approximate the dual problem! The contribution of our paper is to develop a family of probability distribution parametrizations for the dual, obtained by first unfolding the HMC sampling steps and then learning neural network functions in place of these steps. The optimal solution of the coupled model and dual (Section 3.3) is exactly the target distribution. Thus, the neural network "dual sampler" is automatically adjusted to reduce the finite-step approximation error of HMC. This is not true of a fixed HMC sampler. Finally (Theorem 4) the learned "neural HMC" dual has an explicit closed form density estimate (eq. 17), which can in turn be used to evaluate the entropy. To our knowledge, none of the above contributions have appeared elsewhere. We will clarify these points in the final version. We agree that a preconditioner $K$ can be used for the kinetic energy in the HMC unfolding. Indeed, in the generalized HMC layer in Eq.(13), we considered even more general preconditioners, which include not just linear preconditioners but also nonlinear projections to latent low-dimension spaces, with automatically learned parameters.

- **Effect of SGD for inner problem.** As we showed in Theorem 1, the optimal solution to the inner minimisation is the target sampling distribution. Therefore, SGD will *not* "blur the precision of HMC". Instead, solving the inner minimisation by SGD will lead the dual sampler, *i.e.*, the proposed dynamcis embedding network, to the actual target distribution and reduce the finite-step approximation error.
  Theorem 3 justifies the flexibility of the proposed unfolded dynamics-based sampler, saying that as $T \to \infty$, the neural network parametrization for the dual sampler can approach any exponential family density arbitrarily well. This is not related to the effect of SGD in the inner minimisation.

- **Number of sampling steps.** In our synthetic experiments, we have the initial distribution in ADE for $v$ as a standard Gaussian and for $x$ as $q^0(x)$, which is parametrized by a 10-layer normalizing flow, with extra 5 dynamics-based sampling layers. For fairness, we compared to CD with 15 fixed HMC-steps. The experiments demonstrate that given same number of steps, ADE will achieve a smaller approximation error by learning the HMC sampler, resulting in better performance. The leapfrog stepsize is learned in ADE and tuned in CD-HMC following [40]. We tested HMC varying the number of steps as $T = 1, 3, 5, 15$. HMC-15 performs best, as reported in the main text. This is also confirmed by our theory: as $T$ becomes larger, the approximation error becomes smaller, and thus, the performance achieved is better.

[Meta-Review · NeurIPS 2019]

A novel approach to ML learning by taking the dual of the log-partition function, resulting in a saddle-point (or adversarial) problem with kinetic variables. Reviews were consistent that the approach was novel and may spur further research.